# Validation of non-participation bias methodology based on record-linked Finnish register-based health survey data: a protocol paper

Megan A McMinn,[1] Pekka Martikainen,[2] Emma Gorman,[3] Harri Rissanen,[4] Tommi Härkänen,[4] Hanna Tolonen,[4] Alastair H Leyland,[1] Linsay Gray[1]

[1]MRC/CSO Social and Public Health Sciences Unit, University of Glasgow, Glasgow, UK
[2]Population Research Unit, Faculty of Social Science, University of Helsinki, Helsinki, Finland
[3]Department of Economics, Lancaster University, Lancaster, UK
[4]Department of Public Health Solutions, National Institute for Health and Welfare, Helsinki, Finland

**Correspondence to**
Dr Megan A McMinn;
megan.mcminn@glasgow.ac.uk

## ABSTRACT

**Introduction** Decreasing participation levels in health surveys pose a threat to the validity of estimates intended to be representative of their target population. If participants and non-participants differ systematically, the results may be biased. The application of traditional non-response adjustment methods, such as weighting, can fail to correct for such biases, as estimates are typically based on the sociodemographic information available. Therefore, a dedicated methodology to infer on non-participants offers advancement by employing survey data linked to administrative health records, with reference to data on the general population. We aim to validate such a methodology in a register-based setting, where individual-level data on participants and non-participants are available, taking alcohol consumption estimation as the exemplar focus.

**Methods and analysis** We made use of the selected sample of the Health 2000 survey conducted in Finland and a separate register-based sample of the contemporaneous population, with follow-up until 2012. Finland has nationally representative administrative and health registers available for individual-level record linkage to the Health 2000 survey participants and invited non-participants, and the population sample. By comparing the population sample and the participants, synthetic observations representing the non-participants may be generated, as per the developed methodology. We can compare the distribution of the synthetic non-participants with the true distribution from the register data. Multiple imputation was then used to estimate alcohol consumption based on both the actual and synthetic data for non-participants, and the estimates can be compared to evaluate the methodology's performance.

**Ethics and dissemination** Ethical approval and access to the Health 2000 survey data and data from administrative and health registers have been given by the Health 2000 Scientific Advisory Board, Statistics Finland and the National Institute for Health and Welfare. The outputs will include two publications in public health and statistical methodology journals and conference presentations.

## Strengths and limitations of this study

► This study will validate a dedicated methodology that aims to adjust for non-participation bias in health surveys through the use of record linkage.
► We use an individual-level dataset on the entire selected sample for a Finnish national health survey, from which the characteristics of non-participants can be identified, with linkage to morbidity and mortality records, providing the 'gold standard' for the methodology validation process.
► Previous applications of this methodology have been able to use data on the total population for comparison. This study is limited to a population sample available for this analysis.
► The estimated gradient in the risk of alcohol-related harms may be stronger using individual measures of socioeconomic position than area-level measures of deprivation; therefore, these reference comparisons may not mirror the methodology based on less informative area-based measures.
► This validation exercise is confined to assessing the reliability of inferring on non-participants from comparisons of the participants and the reference population; other aspects of the methodology, such as the extent to which alcohol-related hospitalisations and deaths provide sufficient information to impute unknown alcohol consumption estimates, are beyond the scope of this study.

such as smoking prevalence, levels of physical activity and alcohol consumption for entire populations, not confined to the subpopulation in contact with health services. However, the decreasing levels of participation in these surveys threaten their ability to provide reliable estimates.[1–3] The proportions of non-participation are typically not uniform across sociodemographic groups, meaning that selected groups, such as men or those from deprived backgrounds, are often under-represented in health surveys.[4] Non-participation has also been found to

## INTRODUCTION

Health surveys enable the production of estimates of various health-related behaviours,

correlate with higher rates of morbidity and mortality[5 6]; in particular, substantially lower rates of alcohol-related harms (deaths and hospitalisations) have been found among participants, compared with the general population.[7] Where it is possible to identify non-participants, findings of higher harm rates among the non-participants relative to the participants have been reported.[8 9] A set of health studies conducted in Finland found that deaths due to alcohol-related diseases, injuries and poisonings had the largest relative mortality differences between participants and non-participants for men and were second largest for women, exceeded only by deaths due to suicides.[9] In Denmark, non-participants were found to have significantly increased hazard ratios for alcohol-related hospitalisations and deaths relative to participants.[8] Under such circumstances, there is bias present in the participant sample and, as a consequence, in the derived estimates of alcohol consumption. Attempts to correct for such non-participation bias typically make use of weights based on sociodemographic characteristics[10]; however, this may not fully capture health differences. The success of the weighting is dependent on the extent to which those participating are representative of their subgroups of the population. For instance, individuals in harder-to-reach subgroups, such as younger men from disadvantaged backgrounds, that do participate, are unlikely to be representative of their entire demographic, and so weighting does not resolve the bias.[11]

We have developed[11] and applied[6 12] a dedicated methodology that uses additional health information from data linkage and reference to population data to adjust for non-participation bias. This methodology has previously been used to improve estimates of population-level alcohol consumption, although it could be applied to other health-related behaviours of interest, such as tobacco smoking.

Briefly, the methodology makes inference on the non-participants by comparing the sociodemographic characteristics and rates of (in the case of the previous application: alcohol-related) hospitalisations and deaths in the survey participants, to the population, identifying any deviations in representativeness. Any differences point to non-participation in the respective sociodemographic-health grouping. The number of synthetic observations on non-participants to generate in each sociodemographic-health group is based on the number of participants and the overall participation rate, with uncertainty due to sampling variation introduced through the use of repeated bootstrap samples and random rounding. Multiple imputation is then used to fill in values for the 'missing' variables collected in the health survey (alcohol consumption, in the case of the application) for these 'non-participants'. Multiple imputation has the flexibility to accommodate differences between participants and non-participants within groups defined by harm status as well as sociodemographic characteristics. Application of this methodology in Scotland found that mean weekly alcohol consumption was between 14% and 53% higher for men after non-response bias was corrected for, depending on

how extreme the differences in sex-specific mean weekly consumption between participants and non-participants were assumed to be, with little impact on estimates for women.[12]

This project aims to validate the methodology developed for addressing non-participation bias. More specifically, to evaluate whether it is valid to infer on the non-participants from comparisons of the participants and a total register-based population sample without non-response. Validation requires a setting whereby some true information on the individual non-participants of a health survey is known, and these can be compared with the synthetic observations generated by our methodology. Finland provides this opportunity as it maintains a nationally representative register that forms the sampling frame for surveys and has the ability to interlink sociodemographic information, morbidity and mortality databases, and survey responses at the individual level using personal identification codes.[13] Therefore, through the use of this register, the sociodemographic, hospitalisation and death categories of the true non-participants are known (providing the 'gold standard'). With the addition of the general population data, we are able to make indirect inference using the synthetic observations. We can then compare the results of the synthetic and true non-participants, allowing us to assess the validity of our existing methodology.

## METHODS AND ANALYSIS
### Health 2000 survey data
The Health 2000 Survey (thl.fi/health2000) is a nationally representative health examination survey conducted in Finland between 2000 and 2001. A regional two-stage stratified cluster sampling strategy was used to identify approximately 8000 persons aged 30 and over, in the main survey.[14] Figure 1 describes the Health 2000 sample, and the process of identifying the subsample for this analysis. The sample members aged 30 and over (n=8028) were invited to participate in a home-based interview, to self-complete a health questionnaire and to attend a health examination. The health questionnaire[15] comprised questions related to living habits and environments, and included questions on the types, quantities and frequency of alcohol consumed over the past 12 months, from which we can derive an estimate of average weekly alcohol consumption, measured in grams per week. Of the persons aged 30 and over, 84.4% (n=6736) completed the health questionnaire. These were considered to be participants for the purposes of this validation project, as the health questionnaire was the source of the outcome of interest—average weekly alcohol consumption. For the purposes of our study, non-participants comprised those who had not completed the health questionnaire, as well as those who had not participated in any part of the survey, resulting in a total of 1243 (15.6%) non-participants. Given that multiple comorbidities are more common at the older ages, and advanced ages are likely to have a lowering tolerance to alcohol and a change in

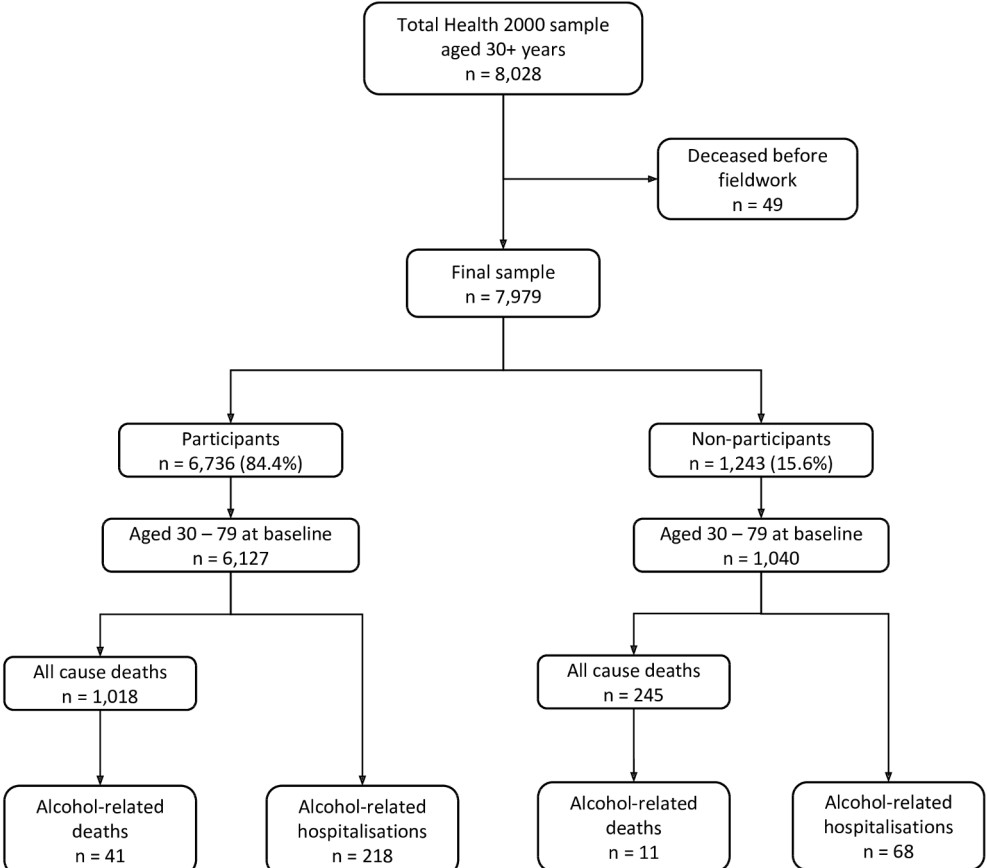

**Figure 1** Analytic sample selection process. Deaths and hospitalisations on 31 December 2015.

drinking patterns,[16 17] the subsample for this analysis, described in figure 1, will be limited to those aged 30–79 years at the start of follow-up.

The selected Health 2000 sample was drawn from the Population Register Centre dataset, held by the Social Insurance Institution (Kela) of Finland.[18] The outcome of interest, average weekly alcohol consumption, is derived using the self-reported frequency and quantity of three types (beer, wine and spirits) of alcohol consumed in the past month/12 months, collected in Health Questionnaire 1.

Sampling weights were calculated by the Health 2000 study team and were estimated based on a design weight, health centre district and university hospital district indicators, 10-year age group, gender and native language (Finnish or Swedish). The design weights took the sampling design into account, including stratum and cluster-specific inclusion probabilities, and the oversampling for the those aged over 80.[19] Sampling weights were estimated for persons who had participated in at least one stage of data collection, including home interviews, health exams and questionnaires. Therefore, weights were available for some non-participants, as defined in this analysis, as they had participated in the home interview, but not the questionnaire collecting alcohol consumption, in addition to all participants. The weights will be retained for the participants, and all non-participants will have their weights set to the default value of 1.

### General population data

An 11% sample of the contemporaneous total Finnish population aged 15 years and older, permanently living in Finland at the end of any of the years between 1987 and 2007, was constructed by Statistics Finland, and is available as the reference comparator for this analysis. This sample was supplemented with an additional 80% oversample of deaths occurring in 1988–2007. In line with the age limits used for the Health 2000 sample, the population sample was restricted to those aged 30–79 years alive on 20 October 2000 (median baseline date for Health 2000 survey cohort). Records of all alcohol-related hospitalisations and all-cause deaths occurring from 20 October 2000 to the end of 2012 were individually linked. To negate the wait involved in applying for unnecessary individual-level data, sociodemographic-specific counts of alcohol-related harms and all-cause deaths were provided to us by Statistics Finland and the National Institute for Health and Welfare. These counts were weighted to account for the different sampling probabilities and the oversample of deaths.

### Linked Health 2000: deaths/hospitalisations and educational attainment

In Finland, nationally representative administrative registers, as for other Nordic countries, enables the linkage of both participants and non-participants of the Health 2000 survey, and the 11% sample of

the contemporaneous population to hospitalisation (alcohol related) and death (all causes and alcohol related) records, as well as sociodemographic variables. Educational attainment, dichotomised into four groups (basic, secondary, tertiary and postgraduate), is available for the analysis, along with age, sex and region of residence. Educational attainment for both the Health 2000 sample and the population sample has been sourced from Statistics Finland, ensuring that they are measured at approximately the same time across the population. Linkage provides information on alcohol-related in-patient hospitalisations (date of event and International Classification of Diseases codes) and all-cause and alcohol-related deaths (date of death and ICD codes) from which an indicator of alcohol-related harm can be derived. Follow-up for hospitalisations and deaths of the Health 2000 sample is available until 31 December 2015, whereas follow-up is limited to the end of 2012 for the general population sample. Therefore, to ensure like-for-like comparisons are made, follow-up for all analyses will be truncated at 31 December 2012.

## Statistical methodology

We aim to examine the differences in estimated average weekly alcohol consumption determined from the two methods: basing the alcohol consumption imputation on the actual sociodemographic and harm data at the individual level on the true non-participants versus basing the imputation on the synthetic observations on non-participants generated using the general population (figure 2).

In doing this we will:

1. Quantify the differences in alcohol-related harm and all-cause mortality between survey participants and non-participants, and survey participants and the population sample using rate ratios of alcohol-related harms using Poisson or negative binomial regression.
2. Generate estimates of alcohol consumption by the two approaches:
   a. Perform multiple imputation to estimate values of average weekly alcohol consumption for true non-participants. This imputation will use the known age group, sex, educational attainment, deaths and hospitalisations due to alcohol for the participants and non-participants, obtained from administrative

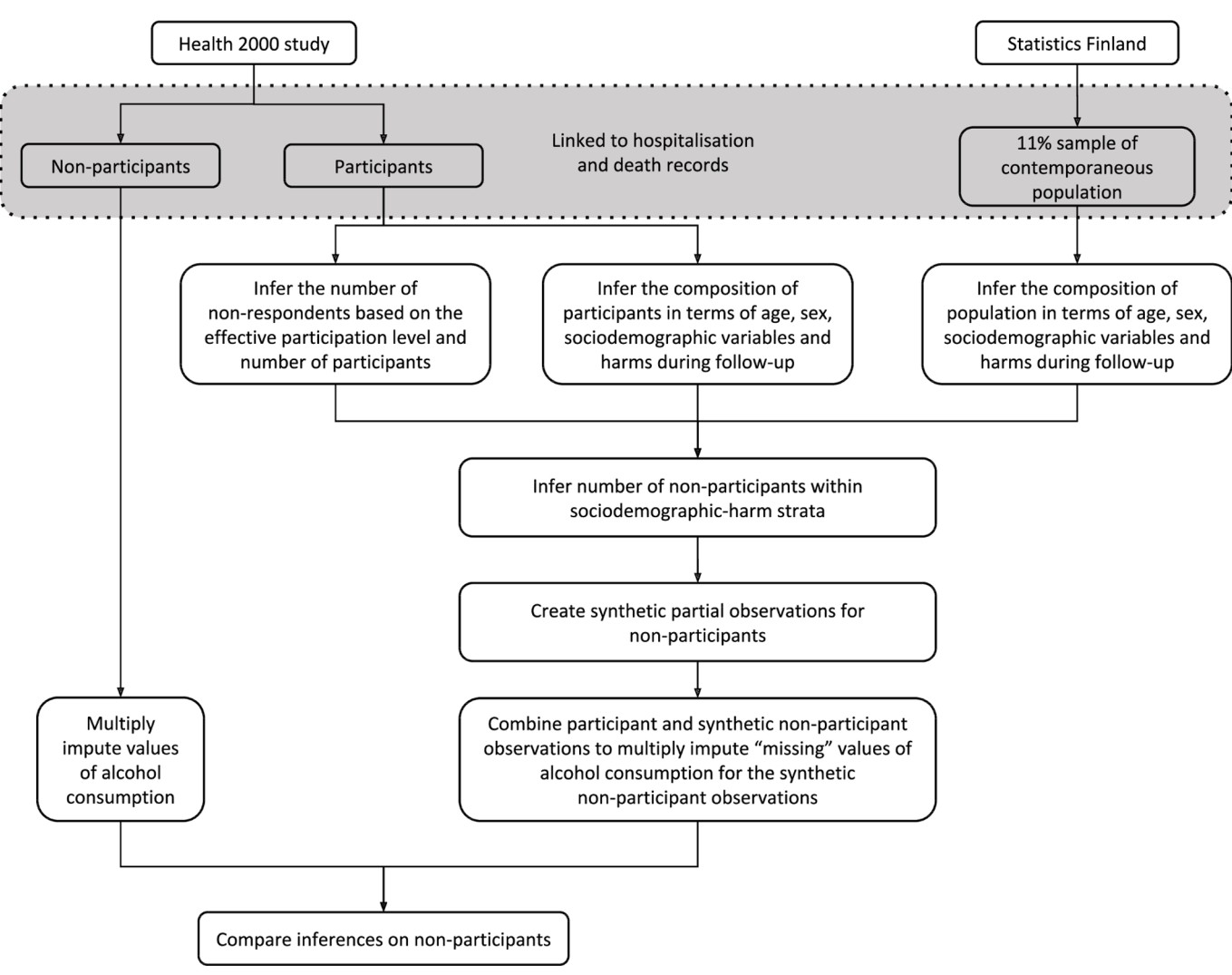

**Figure 2** Summary of the proposed methodology.

data on the total sampling frame, and the known alcohol consumption of the participants alone, obtained from the Health 2000 survey. This provides the gold standard.

 b. Apply the existing methodology outlined in Gray *et al*[11] and executed in Gorman *et al*[12] to create the synthetic observations of sociodemographic measures, and rates of hospitalisation and deaths of non-participants based on a comparison of participants and a contemporaneous total register-based population sample without non-response. Multiple imputation is used to estimate these inferred non-participants' alcohol consumption.

3. Examine how the estimates of alcohol consumption differ between the approaches taken in steps 2a and 2b. This will be measured using the relative difference in mean weekly alcohol estimates, using the gold standard as the reference. Differences will be assessed overall, by sex, and by educational attainment. Repeated bootstrap samples will be used to generate a 95% CI surrounding each relative difference, in order to assess how similar the two approaches are. The proportions of true and inferred non-participants within each age–sex–education–harm-mortality group can similarly be compared with assess how successfully the simulated observations on non-participants reflect the true non-participants.

## IMPLICATIONS

This work has implications for the conduct and analysis of population-sampled surveys. Should the estimated average weekly alcohol consumptions from the two approaches be similar, that is, if the relative difference is smaller than the minimum acceptability limit of 5%, we would consider the methodology a useful tool for correcting bias. The bootstrapped 95% confidence intervals provide guidance as to the statistical significance of the difference. The existing methodology could then be applied to correct for bias arising from non-participation with greater confidence to a wide range of population health measures obtained through health surveys, such as tobacco smoking or physical activity. Should the estimated consumptions differ between the two approaches, further investigations will be required, such as comparisons of the selected survey sample (participants and non-participants) with the population sample.

## Practical/operational issues

The outcome of interest in this project, average alcohol consumed per week, is derived from self-reported drinking status (current, ex and never) and amounts of alcohol consumed by type (beer, wine and spirits). There are several instances where the responses provided conflict between questions, such as those who describe themselves as non-drinkers but also report consuming alcohol within the last 12 months. Average weekly alcohol consumption was calculated by the Health 2000 project team, and this

analysis follows the rules defined in their calculations. A sensitivity analysis will be performed exploring the effects of amending the drinking status and/or average amount consumed in conflicting cases.

The previous applications of this methodology[6 12] were conducted in a setting in which it was natural to use an area-level deprivation index as the socioeconomic measure. No official measure of area-level deprivation exists for Finland, which leads us to using educational attainment for the validation exercise. Given that individual-level measures of socioeconomic position are likely to be more informative than area-based measures, and that relationships between consumption and alcohol-related harms have been found to be stronger using individual-level measures,[20] the application of this methodology to settings with area-based measures may require further validation.

## Ethics and dissemination

The plans and protocol for the Health 2000 survey were reviewed by the National Public Health Institute's Ethical Committee in 1999 and approved by the Ethical Committee for Research in Epidemiology and Public Health at the Hospital District of Helsinki and Uusimaa (HUS) in 2000.

The members in the Health 2000 survey cohort were sent information letters prior to participating in the survey, which included a description of the study contents, the rights of the participants, and the possibility of later linkage to register data. Signed informed consent forms were required from all participants.[14] In Finland, survey data can be linked to registers if (a) survey participants have provided informed consent for this and (b) the register owner provides the right for use of register data.[13] From survey non-participants no survey data exist, so linkage can be done with the permission from the register owner only.

In order to access Health 2000 files for secondary data analysis, as is being performed in this project, researchers were required to submit research plans for approval by the Health 2000 Scientific Advisory Board. Statistics Finland approved access to records of deaths and sociodemographic data, and the National Institute for Health and Welfare (THL) for hospitalisation data of this sample.

The outputs of the research will include two papers: the mortality differences between survey participants and non-participants (step 1), and a comparison of the inference from the two methods (steps 2 and 3).

## Beneficiaries and target audiences

Research on alcohol consumption, and more broadly, methods to improve on estimates derived from health surveys, will be of interest to a range of both academic and non-academic audiences, including users of survey data, epidemiologists, public health and policy researchers, and governmental organisations. The findings of this validation exercise will have implications for general survey conduct: particularly, if the methodology is shown to be

invalid, consideration could be given to basing sampling frames on sources that readily identify non-participants as well as participants and enable linkage to administrative records at the individual level. Should the results of the future analysis demonstrate the validity of the methodology, the approach will be of benefit in the evaluation and creation of public health policy in both local and international governments.

### Patient and public involvement

In this research, data from the general population, not on patients, were used. This analysis utilised two large pseudonymised record-linked administrative datasets with no possibility of direct participant contact beyond their initial participation in the Health 2000 study, due to data protection restrictions. Participants were not invited to contribute to the writing or editing of this document for readability or accuracy.

**Acknowledgements**  We would like to thank the participants of the Health 2000 study, National Institute for Health and Welfare (THL) and Statistics Finland for the provision of the sociodemographic, hospitalisation and death data. Thanks in particular to Joonas JJ Pitkänen from the Population Research Unit, Faculty of Social Sciences at the University of Helsinki, for the preparation and provision of the population data.

**Contributors**  MAM prepared the first draft of this paper. The validation exercise was conceived by LG and AHL in discussion with PM. EG, TH and HR facilitated the acquisition of the survey and register-based data; PM facilitated the acquisition of the population sample data. HT contributed to the formulation of the approach. All authors contributed to all sections of the manuscript and approved the final version.

**Funding**  MAM, LG and AHL receive core funding from the Medical Research Council (MRC_MC_UU_12017/13) and the Scottish Government Chief Scientist Office (SPHSU13). TH, HR and HT are supported by the Academy of Finland under Grant 266251. PM is funded by the Academy of Finland.

**Competing interests**  AHL reports grants from the Medical Research Council and the Chief Scientist Office during the conduct of the study.

**Patient consent for publication**  Not required.

**Ethics approval**  Research plans for this project have been approval by the Health 2000 Scientific Advisory Board. Access to register-based sociodemographic variables and death data has been granted by Statistics Finland and to hospitalisation data by the National Institute for Health and Welfare (THL).

**Provenance and peer review**  Not commissioned; externally peer reviewed.

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
