## [Reviewer comments · BMJ Open]

ARTICLE DETAILS

TITLE (PROVISIONAL)	Validation of non-participation bias methodology based on record-linked Finnish register-based health survey data: a protocol paper
AUTHORS	McMinn, Megan; Martikainen, Pekka; Gorman, Emma; Rissanen, Harri; Härkänen, Tommi; Tolonen, Hanna; Leyland, Alastair; Gray, Lindsay

VERSION 1 - REVIEW

REVIEWER	Dr Gareth R Davies Principal Project Manager Hywel Dda University Health Board Wales, UK
REVIEW RETURNED	03-Oct-2018

GENERAL COMMENTS	This looks like a very interesting and worthwhile study and I only have two minor questions: 1) Methods and analysis - General population data - p7, lines 44-49: you seem to suggest here that you will be using [truncated to 30-79] age standardised rates. As all age standardised rates are artificial constructs I didn't read anything that suggested this was a better approach than using the respective age specific rates. You could have age standardised rates of identical value but of very different composition and not realise it, whereas the age specific rates are always transparent. Please justify the use of age standardised rates here. 2) Methods and analysis - Implications - p7, lines 19-24: you suggest here that the findings relating to alcohol could be used to correct for bias in other health survey questions. I'm unsure whether alcohol can be used a universal measure, for example might non response to illegal issues (drug use) be different?
--

REVIEWER	Sean Randall Curtin University
REVIEW RETURNED	26-Oct-2018

GENERAL COMMENTS	A well written protocol. I have one concern The authors need to discuss how they will distinguish whether their method works or not. They mention they will check to see whether the results differ, but what does 'differ' here mean? Obviously the
--

	results wont be identical. What difference would be small enough that the authors interpret it as being negligible (and thus the new method works fine) and what difference would be large enough that the new method would be deemed unsatisfactory?
--	---

REVIEWER	Jonathan Burton ISER, University of Essex. UK.
REVIEW RETURNED	01-Nov-2018

GENERAL COMMENTS	This is a clear and concise protocol for an interesting research project which uses an 11% sample of the population register to make inferences about the potential response bias in a health study.
--

REVIEWER	Jonas Björk Lund University, Sweden
REVIEW RETURNED	25-Nov-2018

GENERAL COMMENTS	This a well-written protocol on a study with interest from a methodological point of view. I have the following suggestions and comments for the authors: Major issues:  1. I lack a specification of the authors' expectations from a more theoretical point of view. Under what conditions can the models be expected to yield similar results? When will they differ? When will both models be wrong? 2. The plan is to use only health data from the survey time point and onwards, which make both methods less useful for assessment of selection issues in newly established survey cohorts. Data on health care usage and disease history prior to the survey time point may add useful information and will add applicability to the methods. A recent study from Sweden used both population and health care data, retrospectively collected from baseline and backwards, to estimate propensity of participation in a general-population cohort (see Björk et al. Scandinavian Journal of Public Health 2017). 3. It is mentioned in the introduction (and referred to in the abstract) that data on sociodemographic variables, hospitalisation and death categories of the true non-participants are known and will be used as "gold standard". As I understand it, the validation (e.g. against actual socioeconomic situation) is the first step before moving on to the comparison of weekly alcohol consumption. In the Methods-section ("Statistical methodology"), however, this validation against the gold standard is not mentioned explicitly. 4. It is mentioned that no official measure of area level deprivation exists for Finland. Would it be possible to create such an index based on the data that will be available for the present study? Minor issues:
---

	1. It is not clear to me why not all non-participants have sampling weights, as the weights were based on population register data (page 7). Nor is it clear to me why the non-participants will have their sample weights set to one. Please clarify these issues. 2. The text is a bit repetitive regarding the inclusion criteria (page 7 vs. page 8) 3. A useful reference in this context is Stuart et al. (Prev Sci 2015), which describes how to use inverse probability weighting to correct for selection bias. 4. The text would benefit from a somewhat longer description of the method by Gray et al. (2013), which is essential for the study. The paper would also benefit from a short description of the socioeconomic groupings that are going to be used.
--	--

VERSION 1 – AUTHOR RESPONSE

Our response to each reviewer's comments follows

1. Methods and analysis - General population data - p7, lines 44-49: you seem to suggest here that you will be using [truncated to 30-79] age standardised rates. As all age standardised rates are artificial constructs I didn't read anything that suggested this was a better approach than using the respective age specific rates. You could have age standardised rates of identical value but of very different composition and not realise it, whereas the age specific rates are always transparent. Please justify the use of age standardised rates here.

Response: Thank you for identifying this mistake – sociodemographic-specific counts of alcohol-related harms and all cause deaths were provided to us, not rates. Further detail has also been provided on the construction of the population sample (p 6-7).

2. Methods and analysis - Implications - p7, lines 19-24: you suggest here that the findings relating to alcohol could be used to correct for bias in other health survey questions. I'm unsure whether alcohol can be used a universal measure, for example might non response to illegal issues (drug use) be different?

Response: We have amended the text to state that the methodology, rather than the results of the alcohol consumption itself, could be used to correct for bias in other measures. (p. 8)

3. The authors need to discuss how they will distinguish whether their method works or not. They mention they will check to see whether the results differ, but what does 'differ' here mean? Obviously the results won't be identical. What difference would be small enough that the authors interpret it as being negligible (and thus the new method works fine) and what difference would be large enough that the new method would be deemed unsatisfactory?

And

4. I lack a specification of the authors' expectations from a more theoretical point of view. Under what conditions can the models be expected to yield similar results? When will they differ? When will both models be wrong?

Response: We have included further detail on how we will evaluate the success of the validation exercise on p. 8. In particular, we will calculate the relative difference between the mean weekly alcohol consumption estimates of the synthetic and 'gold standard' samples, using the gold standard as the reference. The similarity between the two will be assessed using repeated bootstrap samples to generate a 95% confidence interval surrounding the relative difference. The methodology will be deemed validated if the 95% confidence interval contains the null difference.

5. The plan is to use only health data from the survey time point and onwards, which make both methods less useful for assessment of selection issues in newly established survey cohorts. Data on health care usage and disease history prior to the survey time point may add useful information and will add applicability to the methods. A recent study from Sweden used both population and health care data, retrospectively collected from baseline and backwards, to estimate propensity of participation in a general-population cohort (see Björk et al. *Scandinavian Journal of Public Health* 2017).

Response: Thank you for this suggestion. This protocol details a validation exercise of the methodology as it stands currently, and therefore we are limiting the data used in this analysis to what was available in the initial development of the methodology in Scotland.

However, this suggestion is planned for future advancements of the methodology.

6. It is mentioned in the introduction (and referred to in the abstract) that data on sociodemographic variables, hospitalisation and death categories of the true non-participants are known and will be used as "gold standard". As I understand it, the validation (e.g. against actual socioeconomic situation) is the first step before moving on to the comparison of weekly alcohol consumption. In the Methods-section ("Statistical methodology"), however, this validation against the gold standard is not mentioned explicitly.

Response: Thank you for identifying this oversight. The 'gold standard' is estimated in step 2a (multiple imputation to estimate the average weekly alcohol consumption of the true non-participants). We have now explicitly stated that this step is the 'gold standard' on p. 8.

7. It is mentioned that no official measure of area level deprivation exists for Finland. Would it be possible to create such an index based on the data that will be available for the present study?

Response: Whilst area based measures could theoretically be constructed (such as the Carstairs indices used in Scotland), it is beyond the scope of this project, given they would require extensive validation for the Finnish setting to ensure that the components of the score are theoretically valid measures of deprivation in Finland.

8. It is not clear to me why not all non-participants have sampling weights, as the weights were based on population register data (page 7). Nor is it clear to me why the non-participants will have their sample weights set to one. Please clarify these issues.

Response: Thank you for identifying this ambiguity. We have rephrased the section explaining the weights available for a small number of non-participants on p. 6.

The sampling weights were calculated by the Health 2000 project team in order for the sample participants to be able to be generalizable to the target population. Sampling weights were therefore not estimated for those who did not participate in any part of the study. These individuals have a missing value for their weight in the survey dataset.

In this analysis, some of the individuals who participated did not complete the questionnaire pertaining to alcohol consumption. Therefore they will still have sampling weights calculated for them by the Project Team, but would be regarded as a non-participant in the analysis.

For all analyses the weights of non-participants were set to the default value of 1 (replacing calculated weights and missing values), rather than set to missing. This ensures they would be included in the regression and imputation models, rather than disregarded due to their missingness.

9. The text is a bit repetitive regarding the inclusion criteria (page 7 vs. page 8)

Response: Thank you for identifying this. As the inclusion criteria were detailed in earlier sections, the Inclusion Criteria section has been removed.

10. A useful reference in this context is Stuart et al. (Prev Sci 2015), which describes how to use inverse probability weighting to correct for selection bias.

Response: Thank you for suggesting this paper, it fed into the development of our thinking on how to measure the similarity between the two approaches.

11. The text would benefit from a somewhat longer description of the method by Gray et al. (2013), which is essential for the study. The paper would also benefit from a short description of the socioeconomic groupings that are going to be used.

Response: As recommended, further detail of the method is included on pages 4 and 5. As explained in the introduction to this review, we have replaced our socioeconomic grouping variable with educational attainment. The four categories available for analysis have been listed on page 7.

VERSION 2 – REVIEW

REVIEWER	Dr Gareth R Davies Hywel Dda University Health Board, Wales, UK.
REVIEW RETURNED	12-Jan-2019

GENERAL COMMENTS	Thank you for satisfactorily addressing my two original concerns (1-2), best wishes for your research.
--

REVIEWER	Sean Randall Curtin University, Australia
REVIEW RETURNED	25-Jan-2019

GENERAL COMMENTS	The authors have addressed my comment, outlining that they will view their method as comparable based on finding no statistically significance difference. This is standard practice; however I would encourage the authors to also think about effect sizes - even the smallest difference will be statistically significant with enough numbers, and we can never actually expect the difference to be truly zero.
--

REVIEWER	Jonas Björk Lund University, Sweden
REVIEW RETURNED	19-Jan-2019

GENERAL COMMENTS	The authors have responded adequately to most of my comments on the previous version, and have also incorporated adequate changes. I have three remaining comments: 1. I still lack the theoretical justification for the work. It must be possible to work out under which conditions the two methods can be expected to give the same correct result, same but incorrect results and diverging results (e.g. one correct, one incorrect). Without this theoretical background, it will be hard, if not impossible, to draw firm conclusions from the suggested empirical study. 2. The claimed "gold standard" is obtained by multiple imputation. Thus, we will only know whether two methods yield similar results or not in this particular setting (but they can both be wrong). Since Finnish register data are used, I would suggest to include some register-based health variables as an additional (more gold-like) reference. The advantage with this would be that such variables are available for everyone, participants as well as non-participants. A recent study (see Bonander et al., Jrn of Clin Epi 2019, e-pub) uses this approach. 3. Inspecting whether the 95% confidence interval for the relative difference contains zero is a weak criterion for assessing representativeness, as "absence of evidence is not necessarily evidence of absence". I would suggest that the interpretation of the results is based on the estimated difference, where the 95% confidence interval should serve as guidance as to how statistically certain the conclusion is.
---

VERSION 2 – AUTHOR RESPONSE

We would like to thank the three reviewers for their additional comments on this manuscript.

Our response to each reviewer's comments follow:

1. I still lack the theoretical justification for the work. It must be possible to work out under which conditions the two methods can be expected to give the same correct result, same but incorrect results and diverging results (e.g. one correct, one incorrect). Without this theoretical background, it will be hard, if not impossible, to draw firm conclusions from the suggested empirical study.

This project aims to evaluate whether it is valid to infer on non-participants from comparisons of the participants and a total register based population sample. This requires two stages:

(a) simulate the non-participants (age, sex, education, experience of alcohol-related harm and all-cause death), based on the comparisons of the participants and the population sample; and

(b) multiply impute the alcohol consumption of true and inferred non-participants.

For (a) we have a "gold standard" – the distribution of true non-participants within each of the age/sex/education/harm/mortality strata. If the simulated observations match the true non-participants, then they are the same, and give the correct result. Diverging results would indicate a failure in the simulation of the non-participants. The situation of the 'same but incorrect results' is not possible, given the gold standard is known.

For (b) we don't have gold standard data (here this would be alcohol sales data, broken down by age/sex/education), so it could be possible that the estimated consumption for both could be 'incorrect'. The aim of this work, however, is to compare the estimates from the two methods, and not claim that we have estimated the 'true' consumption, as stated in the strengths and limitations of the article summary section.

We have made the comparisons of the inferred and true non-participant distributions more explicit in step 3 of the methodology section.

2. The claimed "gold standard" is obtained by multiple imputation. Thus, we will only know whether two methods yield similar results or not in this particular setting (but they can both be wrong). Since Finnish register data are used, I would suggest to include some register-based health variables as an additional (more gold-like) reference. The advantage with this would be that such variables are available for everyone, participants as well as non-participants. A recent study (see Bonander et al., *Jrn of Clin Epi* 2019, e-pub) uses this approach.

As detailed in the reply above – the gold standard is not obtained by multiple imputation. The gold standard data contain already contain indicators for alcohol-related harm and mortality of the true non-participants, in addition to their age, sex and educational attainment at baseline.

Therefore, we are still able to assess how well the simulated observations on non-participants reflect the true distributions of the non-participants from the gold-standard sample (in age/sex/education/harm/mortality strata).

3. Inspecting whether the 95% confidence interval for the relative difference contains zero is a weak criterion for assessing representativeness, as "absence of evidence is not necessarily evidence of absence". I would suggest that the interpretation of the results is based on the estimated difference, where the 95% confidence interval should serve as guidance as to how statistically certain the conclusion is.

and

4. The authors have addressed my comment, outlining that they will view their method as comparable based on finding no statistically significance difference. This is standard practice; however I would encourage the authors to also think about effect sizes - even the smallest difference will be statistically significant with enough numbers, and we can never actually expect the difference to be truly zero.

In light of these comments, we have revised the validity measure to focus mainly on the size of the relative differences (page 8). In particular, we would consider the methods to agree if the relative differences are less than 5%, with the bootstrapped 95% confidence intervals providing guidance for statistical significance.